# Ninjurin 2, a Cell Adhesion Molecule and a Target of p53, Modulates Wild-Type p53 in Growth Suppression and Mutant p53 in Growth Promotion

**DOI:** 10.3390/cancers16010229

**Published:** 2024-01-04

**Authors:** Jin Zhang, Xiangmudong Kong, Hee Jung Yang, Shakur Mohibi, Christopher August Lucchesi, Weici Zhang, Xinbin Chen

**Affiliations:** 1Comparative Oncology Laboratory, The University of California, Davis, CA 95616, USA; dkkong@ucdavis.edu (X.K.); hjyang3096@lgchem.com (H.J.Y.); smohibi@ucdavis.edu (S.M.); calucchesi@ucdavis.edu (C.A.L.); 2Division of Rheumatology, Allergy and Clinical Immunology, The University of California, Davis, CA 95616, USA; ddzhang@ucdavis.edu

**Keywords:** NINJ2, p53, cell proliferation, cellular senescence

## Abstract

**Simple Summary:**

The nerve injury-induced protein 2 (NINJ2) is a cell adhesion molecule, but its biological function remains largely unexplored. Here, we show that NINJ2 can be induced by tumor suppressor p53. Interestingly, we also found that Ninj2 can in turn modulate p53 expression via repressing both wild-type and mutant p53 mRNA translation. Consequently, we found that the loss of NINJ2 inhibits cell growth in wild-type p53-containing cells but promotes cell growth in mutant p53-containing cells. Together, our data reveal a novel feedback loop between NINJ2 and p53 and that NINJ2 exerts an opposing role in cell growth depending on the p53 status.

**Abstract:**

The nerve injury-induced protein 1 (NINJ1) and NINJ2 constitute a family of homophilic adhesion molecules and are involved in nerve regeneration. Previously, we showed that NINJ1 and p53 are mutually regulated and the NINJ1-p53 loop plays a critical role in p53-dependent tumor suppression. However, the biology of NINJ2 has not been well-explored. By using multiple in vitro cell lines and genetically engineered mouse embryo fibroblasts (MEFs), we showed that *NINJ2* is induced by DNA damage in a p53-dependent manner. Moreover, we found that the loss of NINJ2 promotes p53 expression via mRNA translation and leads to growth suppression in wild-type p53-expressing MCF7 and Molt4 cells and premature senescence in MEFs in a wild-type p53-dependent manner. Interestingly, NINJ2 also regulates mutant p53 expression, and the loss of NINJ2 promotes cell growth and migration in mutant p53-expressing MIA-PaCa2 cells. Together, these data indicate that the mutual regulation between NINJ2 and p53 represents a negative feedback loop, and the NINJ2-p53 loop has opposing functions in wild-type p53-dependent growth suppression and mutant p53-dependent growth promotion.

## 1. Introduction

The tumor suppressor p53 is a sequence-specific transcription factor and known to inhibit cancer formation via maintaining genome integrity [1,2,3]. In response to DNA damage and other stress signals, p53 is activated and then induces the expression of various downstream targets associated with cell cycle arrest (p21, GADD45) [4,5], apoptosis (PUMA, IGFBP3, DR5) [6,7,8], senescence (PAI-1, DEC1) [9,10] and many others. Owing to these activities, p53 is often referred to as the “guardian of the genome”. The importance of p53 in tumor suppression is underscored by the fact that p53 mutation occurs in more than half of human cancers. Indeed, the vast majority of tumor-derived TP53 mutations occur in the region encoding p53’s DNA binding domain, consistent with the role of p53-mediated transcription in tumor suppression [11,12]. Notably, in addition to the loss of the tumor-suppressive function of wild-type (WT) p53, many mutant p53 proteins acquire new oncogenic activities to promote cancer progression, called gain of function (GOF) [13,14,15]. Thus, the unique characteristics of WT and mutant p53 necessitate a consideration of p53 status when targeting p53 for cancer management.

Despite its crucial role in tumor suppression, WT p53 protein is widely considered undruggable owing to its function as a transcription factor. Thus, the current strategies are set to target p53 regulators or TP53 mutation in cancer. One of the most advanced efforts to target p53 for cancer therapy involves efforts to inhibit MDM2 in tumors harboring WT p53, which leads to the development of Nutlin-3 and peptide inhibitors of MDM2 and MDMX [16,17]. Another attractive therapeutic approach involves identifying agents that convert mutant p53 to regain WT p53 activity for tumor suppression [18,19,20]. Nevertheless, the ongoing challenge in the field of p53 research lies in effectively translating the findings from extensive studies into clinical applications.

The nerve injury-induced protein 1 and 2 (Ninjurin 1/2, NINJ1/2) are double-transmembrane cell surface proteins and constitute the Ninjurin family of homophilic adhesion molecules. Both NINJ1 and NINJ2 were originally identified to be induced by nerve injury and promote axonal growth [21,22,23]. Structurally, NINJ1 and NINJ2, which are 52% identical and 67% similar in amino acid sequence, consist of two hydrophobic transmembrane domains, an intracellular domain, and two extracellular domains. Despite sharing structural similarities, NINJ1 and NINJ2 exhibit distinct expression profiles. NINJ1 is ubiquitously expressed in epithelial tissues, such as breast, liver and spleen [22], whereas NINJ2 is highly expressed in hematopoietic and lymphatic tissues [23]. Moreover, recent studies have shown that NINJ1 participates in diverse physiological processes, such as nerve regeneration, inflammation, tumorigenesis, and tissue homeostasis [24,25,26,27,28,29]. Previous studies from our group showed that NINJ1 is a target of p53 and mediates p53-dependent tumor suppression [30,31]. We also found that a small peptide derived from N-terminal extracellular domain was able to induce p53 expression and subsequently, p53-dependent growth suppression [30], suggesting that NINJ1 may be explored as a target for cancer management. However, very little is known about the biological function of NINJ2, and even less is known about its role in tumorigenesis.

In this study, we showed that *NINJ2* is induced by p53, whereas p53 expression is regulated by NINJ2 via mRNA translation. We also showed that NINJ2-p53 loop has opposing functions in WT p53-dependent growth suppression and mutant p53-dependent growth promotion.

## 2. Material and Methods

### 2.1. Reagents 

Anti-p53 (Do-1) (to detect human p53) and anti-p21 were purchased from Santa Cruz Biotechnology (Santa Cruz, CA, USA). Anti-p53 (1C12) (to detect murine p53) was purchased from Cell Signaling Technology (Danvers, MA, USA). Anti-actin and proteinase inhibitor cocktail were purchased from Sigma (St. Louis, MO, USA). Anti-Ninj2 antibody was custom-made by Cocalico biologicals Inc (Denver, PA, USA) using a synthetic peptide (PGSSDPRSQPINLNHYATK). RiboLock RNase Inhibitor and Revert Aid First Strand cDNA Synthesis Kit were purchased from Thermo Fisher (Waltham, MA, USA). Magnetic Protein A/G beads were purchased from MedChemExpress (Monmouth Junction, NJ, USA). WesternBright ECL HRP substrate was purchased from Advansta (San Jose, CA, USA). Scrambled siRNA (5′-GGC CGA UUG UCA AAU AAU U-3′), NINJ2 siRNA#1 (5′-UGA AUG AGG UAG AAA AGC AUU-3′) and NINJ2 siRNA#2 (5′-CUG CCA GGG CCU CAA GG AAU U-3′) were purchased from Horizon Discovery (Chicago, IL, USA). For siRNA transfection, the Neon Electroporation System from Life Technologies (Carlsbad, CA, USA) was used according to the user’s manual. 

### 2.2. Plasmids 

To generate a vector expressing a single guide RNA (sgRNA) that targets *NINJ2*, two 25 nt oligos were annealed and then cloned into the pSpCas9(BB) sgRNA expression vector via BbsI site [32]. The guide RNA sequences are 5′-CGC CTT CAG CCG CAT GGC GTT GG-3′ and 5′- TCT TGG TGG CGT AAT GGT TC -3′.

### 2.3. Mice and Isolation of MEFs 

*Ninj2*-deficient mice (on a pure C57BL/6 background) were generated by the Mouse Biology Program at University of California at Davis. *p53*^+/−^ mice (C57BL/6J strain), developed in the laboratory of Dr. Tyler Jacks [33], were obtained from Jackson laboratory. To generate WT, *Ninj2*^+/−^, *Ninj2*^−/−^ MEFs, *Ninj2*^+/−^ mice were bred with *Ninj2*^+/−^ and MEFs were isolated from 12.5 to 13.5 postcoitum (p.c.) mouse embryos as described previously [34]. To generate compound MEFs, *Ninj2*^+/−^ mice were crossed with *p53*^+/−^ mice to generate double heterozygous mice. The latter were intercrossed to generate *p53*^−/−^ and *Ninj2*^−/−^*; p53*^−/−^ MEFs. MEFs were cultured in DMEM supplemented with 10% FBS (Life Technologies), 55 μM β-mercaptoethanol, and 1× non-essential amino acid (NEAA) solution (Life Technologies). The genotyping primers to detect wild-type *Ninj2* allele were a forward primer, 5′-GGC TAT ACA GAG AAA CTC TGC CCT GG-3′, and a reverse primer, 5′-AGC ACT CTC TGA ATA CCC TGG TTG G-3′. The genotyping primers to detect *Ninj2*-KO allele were a forward primer, 5′-GGG ATC TCA TGC TGG AGT TCT TCG-3′, and the same reverse primer for the wild-type allele. All animals and use protocols were approved by the University of California at Davis Institutional Animal Care and Use Committee.

### 2.4. Cell Culture and Cell Line Generation 

MCF7, A549, H1975, and MIA-PaCa2 cells and their derivatives were cultured in DMEM (Dulbecco’s modified Eagle’s medium, Life Technologies) supplemented with 10% fetal bovine serum (Life Technologies). Molt4 cells and their derivatives were cultured in RPMI 1640 (Life Technologies) supplemented with 10% fetal bovine serum (Life Technologies). p53-KO MCF7 cells were generated previously [35]. To generate *NINJ2*-KO cell lines, MCF7, MIA-PaCa2 and Molt4 cells were transfected with pSpCas9(BB)-2A-Puro vector expressing *NINJ2* guide RNA#1 and #2, followed by selection with puromycin for 2–3 weeks. For *NINJ2*-KO MCF7 and MIA-PaCa2 cell lines, each individual clone was picked, whereas for *NINJ2*-KO Molt4 cell line, cells were sorted by FACS. All individual clones were tested for *NINJ2* expression by Western blot analysis, followed by sequence confirmation.

### 2.5. Western Blot Analysis 

Western blot analysis was performed as previously described [36]. Briefly, protein lysates suspended in Laemmli sample buffer were loaded to 10–13% SDS-polyacrylamide gel. Separated proteins were transferred onto a nitrocellulose membrane (Amersham Hybond, GE Healthcare Bio-Sciences AB, Uppsala, Sweden), followed by incubation with a primary and secondary antibody. The membrane was then incubated with WesternBright Sirius HRP substrate (Advansta) to visualize the target protein by using chemi-doc (AnalytikJena, Upland, CA, USA). The VisionWorks^®^LS software (AnalytikJena, version 8.20) was used to quantitate the relative level of proteins. The original western blot figures can be found in Appendix A.

### 2.6. RNA Isolation and RT-PCR 

Total RNA was isolated with Trizol reagent as described according to the user’s manual (Life Technologies). The total RNA was then used to synthesize cDNA by using RevertAid RT Reverse Transcription Kit (ThermoFisher Scientific; Waltham, MA, USA), followed by PCR analysis. The program used for amplification was (i) 94 °C for 5 min, (ii) 94 °C for 45 s, (iii) 58 °C for 45 s, (iv) 72 °C for 30 s, and (v) 72 °C for 10 min. From steps 2 to 4, the cycle was repeated 28–35 times depending on the targets or 22 times for actin and GAPDH. The primers for *NINJ1* were a forward primer, 5′-ACA TCT TCA TCA CGG CCT TC-3′, and a reverse primer, 5′-GGG AAC AGC TGC TGA GAG AC-3′. The primers for *NINJ2* were a forward primer, 5′- AGA AAA GCA GTG GCG ACT CA, and a reverse primer, 5′-CTG GCA GAT CAC GGA GGA AG-3′. The primers for p53 were a forward primer, 5′-GGC CCA CTT CAC CGT ACT AA-3′, and a reverse primer, 5′-GTG GTT TCA AGG CCA GAT GT-3′. The primers for *GAPDH* were a forward primer, 5′-AGC CTC AAG ATC ATC AGC AAT G-3′, and a reverse primer, 5′-ATG GAC TGT GGT CAT GAG TCC TT-3′. The primers for human and mouse *Actin* were a forward primer, 5′-CTG AAG TAC CCC ATC GAG CAC GGC A-3′, and reverse primer, 5′-GGA TAG CAC AGC CTG GAT AGC AAC G-3′.

### 2.7. ChIP Assay 

The ChIP assay was performed as previously described [7]. Briefly, MCF7 cells were mock-treated or treated with doxorubicin for 18 h. Chromatin was cross-linked in 1% formaldehyde in phosphate-buffered saline (PBS), followed by sonication to yield 200 to 1000 bp DNA fragments. The chromatin lysates were then immunoprecipitated with a control IgG or p53 antibody. After reverse cross-linking and phenol–chloroform extraction, DNA fragments were purified, followed by PCR to visualize the enriched DNA fragments. The primers to detect *NINJ2* promoter were a forward primer, 5′-CTC TGC CTG TCT TCC TCC TG-3′, and a reverse primer, 5′-GCA GGG GTA GT TGT ACC TTC G-3′. The primers to detect p21 promoter were a forward primer, 5′-CAG GCT GTG GCT CTG ATT GG-3′, and a reverse primer, 5′-TTC AGA GTA ACA GGC TAA GG-3′.

### 2.8. Colony Formation Assay 

1 × 10^3^ cells were seeded per well in a six-well plate and then cultured for 2 weeks. When visible clones were formed on the plates, the cells were fixed with methanol/glacial acetic acid (7:1) for 20 min at room temperature and then washed 3 times with distilled water. After that, colonies were stained with 0.1% of crystal violet for 20 min at room temperature, washed with distilled water 3 times, and then air-dried.

### 2.9. Tumor Sphere Assay

MCF7 cells were cultured in MammoCult™ (Stemcell, Cambridge, MA, USA) media as per manufacturer’s guidelines. A total of 20,000 cells per well were seeded in 6-well ultra-low adherent plates. One week later, images of representative tumor spheres were taken using a microscope.

### 2.10. Cell Scratch Assay 

2 × 10^5^ cells were seeded in a 6-well plate and allowed to grow for 24 h. The monolayers were wounded by scraping with a P20 micropipette tip and washed two times with PBS. Microscopic images were taken immediately after scraping (0 h) or 24 h later.

### 2.11. SA-β-Gal Staining 

This assay was performed as described previously [34]. Cells were washed with ice-cold PBS 3 times, fixed with fixation buffer (2% formaldehyde, 0.2% glutaraldehyde) for 15 min at room temperature, and then incubated with SA-β-galactosidase staining solution [1 mg/mL 5-bromo-4-chloro-3-indolyl-β-d-galactopyranoside, 40 mM citric acid/sodium phosphate (pH 6.0), 5 mM potassium ferrocyanide, 5 mM potassium ferricyanide, 150 mM NaCl, and 2 mM MgCl_2_] t at 37 °C overnight. Next day, the β-Galactosidase staining solution was removed, and the cells were overlayed with 70% glycerol and stored at 4 °C. The percentage of senescent cells was calculated as a ratio of SA-β-gal-positive cells versus the total number (~500) of cells counted. 

### 2.12. Click-iT Metabolic Labeling 

The Click-iT metabolic labeling reagents were obtained from Life Science Technology and the assay was performed according to the user’s manual. Briefly, 1 × 10^6^ cells were incubated with methionine-free medium for 60 min to deplete methionine, followed by incubating with HPG (L-homopropargylglycine) for 30 min. Cells were lysated and immunoprecipitated with anti-p53 (Do-1) and then subjected to the Click-iT reaction to label the newly synthesized p53 protein with biotin, which was then detected by Western blot analysis with anti-biotin. 

## 3. Results

### 3.1. NINJ2 Is a Target of p53 Tumor Suppressor

We showed previously that NINJ1 is a target and a regulator of p53 [31]. As a member of the Ninjurin family of homophilic adhesion molecules, we speculate that NINJ2 may also be regulated by the p53 pathway. Thus, we examined whether NINJ2 can be induced by DNA damage in a p53-dependent manner. To test this, MCF7 and Molt4 cells were mock-treated or treated with doxorubicin, followed by RT-PCR to measure the level of *NINJ1* and *NINJ2* transcripts. MCF7 is a human breast cancer cell line, whereas Molt4 is human T lymphoblast cell line, both of which are known to contain WT p53. We found that the levels of *NINJ1* and *NINJ2* transcripts were upregulated by doxorubicin in both MCF7 and Molt4 cells (Figure 1A,B, NINJ1 and NINJ2 panels). Additionally, we showed that *NINJ2* was induced by doxorubicin MCF7 cells in a dose-dependent manner (Figure 1C). In contrast, the *NINJ2* transcript was not upregulated by doxorubicin in p53-KO MCF7 cells (Figure 1D), suggesting that p53 is required for the induction of *NINJ2* in response to DNA damage. In line with this, we showed that upon treatment with doxorubicin or camptothecin, the level of p53 protein was increased, accompanied by elevated expression of NINJ2 in Molt4 cells (Figure 1E). 

Next, we searched the *NINJ2* genomic locus and identified a potential p53-responsive element (p53-RE) in the *NINJ2* promoter (Figure 1F). Thus, a chromatin immunoprecipitation (ChIP) assay was performed and showed that endogenous p53 protein was found to bind to the *NINJ2* promoter, which was increased by treatment with doxorubicin (Figure 1G). The binding of p53 protein to p21 and GAPDH promoters was measured as positive and negative controls, respectively (Figure 1G). Together, these data indicate that *NINJ2* is a target of p53.

### 3.2. NINJ2 Regulates p53 Expression via mRNA Translation and the Mutual Regulation between p53 and NINJ2 Represents a Novel Feedback Loop

Several well-defined p53 targets are known to regulate p53 expression, such as Mdm2, which degrades WT p53 via polyubiquitination [37], and RBM38, which inhibits p53 mRNA translation [34]. To determine whether NINJ2 in turn regulates p53, an siRNA against *NINJ2* was designed and transiently transfected into A549 and Molt4 cells, followed by the detection of WT p53. We found that the level of p53 protein was increased in A549 and Molt4 cells upon the knockdown of *NINJ2* (Figure 2A,B). To verify this, the Crispr-Cas9 gene-editing tool was used to ablate the *NINJ2* gene in both Molt4 and MCF7 cells by using two guide RNAs to target exon 2 of the *NINJ2* gene. Sequence analysis indicated a 38 nt out-of-frame deletion in MCF7 cells and 37 nt out-of-frame deletion in Molt4 cells. Similarly, we found that *NINJ2* knockout led to an increased expression of p53 (Figure 2C,D, p53 panels). Next, to determine how p53 expression is regulated by NINJ2-KO, RT-PCR analysis was performed and showed that the level of p53 transcript was not altered by *NINJ2*-KO (Figure 2E), suggesting that p53 expression is regulated by NINJ2 through post-transcriptional mechanisms, such as mRNA translation. Thus, Click-iT metabolic labeling was performed to measure the newly synthesized p53 protein using isogenic control and *NINJ2*-KO Molt4 cells. We found that the level of de novo p53 protein was much higher in *NINJ2*-KO Molt4 cells than that in isogenic control cells (Figure 2F). These data suggest that p53 mRNA translation is regulated by NINJ2, which is similar to the effect of NINJ1 on p53 mRNA translation [31].

### 3.3. The Loss of Ninj2 Leads to Increased Expression of p53 Accompanied by the Induction of p21 in Mouse Embryo Fibroblasts (MEFs)

The regulatory feedback loop between p53 and its targets is known to be conserved across species. Thus, we generated a *Ninj2*-deficient mouse model by using homologous recombination in embryonic stem cells and classical gene-targeting strategies (Figure 3A). Next, a cohort of WT, *Ninj2*^+/−^, and *Ninj2*^−/−^ MEFs was generated by the intercrossing of *Ninj2*^+/−^ mice (Figure 3B). Consistent with observations in isogenic control and *NINJ2*-KO Molt4 and MCF7 cells (Figure 2A–C), the loss of Ninj2 led to a marked increase in p53 in *Ninj2*^−/−^ MEFs as compared to WT MEFs (Figure 3C). Additionally, p21, a target of p53 [38], was induced in *Ninj2*^−/−^ MEFs (Figure 3C), suggesting that the p53 protein increased by the loss of Ninj2 is functional.

### 3.4. The Loss of NINJ2 Increases the Ability of WT p53 to Induce Growth Suppression, Decrease Cell Migration and Promote Cellular Senescence

To determine the cellular functions of NINJ2, 2D colony formation and 3D tumor sphere assays were performed to measure the role of NINJ2 in cell growth, whereas a scratch assay was performed to measure the role of NINJ2 in cell migration. We found that cell growth was inhibited by *NINJ2*-KO in MCF7 cells, as shown by the reduced numbers of colonies and tumor spheres (Figure 4A,B). We also found that cell migration was inhibited by *NINJ2*-KO (Figure 4C). 

To further elucidate the role of p53 in Ninj2-mediated biological functions, we generated a set of MEFs deficient in *p53*, *Ninj2* or both. As expected, p53 was not detectable in both *p53*^−/−^ and *Ninj2*^−/−^; *p53*^−/−^ MEFs but markedly increased in *Ninj2*^−/−^ MEFs (Figure 4D, p53 panel). Next, SA-β-gal staining was performed to measure cellular senescence. We found that the percentage of SA-β-gal-positive cells was markedly increased in *Ninj2*^−/−^ (38.1%) MEFs as compared to that in WT MEFs (9.1%) (Figure 4E). Most importantly, we found that the level of cellular senescence induced by the loss of *Ninj2* was abrogated by the loss of p53 in *Ninj2*^−/−^; *p53*^−/−^ MEFs (Figure 4E). In line with this, we found that p21, a senescence marker, was induced by *Ninj2*-KO, which was almost abrogated by p53-KO (Figure 4D, p21 panel). These data are also consistent with previous reports that p21 is a major mediator of p53-mediated senescence in MEFs [31,39,40]. Together, these data suggest that the loss of *Ninj2* leads to growth suppression and accelerates cellular senescence in a p53-dependent manner.

### 3.5. The Loss of NINJ2 Increases Mutant p53 Expression and Promotes Cell Growth and Migration

Since p53 expression is regulated by NINJ2 via mRNA translation, we reasoned that NINJ2 would regulate both WT and mutant p53 expression. To this end, H1975, a non-small-cell lung cancer cell line that contains mutant p53-R273H, was used to determine whether Ninj2 regulates mutant p53 expression. Indeed, we found that upon Ninj2 knockdown, the level of mutant p53 was increased in H1975 cells (Figure 5A). To verify this, CRISPR was used to generate *NINJ2*-KO MIA-PaCa2 cell lines. MIA-PaCa2 cells express a mutant p53-R248W, a hot-spot mutant known to possess a gain of function [11]. We showed that upon knockdown of *NINJ2*, mutant p53-R248W was highly expressed as compared to that in isogenic control cells (Figure 5B), consistent with the finding that WT p53 is regulated by NINJ2 in MCF7 and Molt4 cells (Figure 2).

To determine whether NINJ2 would have a biological effect in mutant p53-expressing cells, colony formation and cell migration assays were performed. We showed that the loss of NINJ2 led to increased cell growth as measured by the colony formation assay and increased cell migration by the wound healing assay (Figure 5C,D). The results are opposite from the observations that cell growth and migration were inhibited, whereas cellular senescence was increased by the loss of NINJ2 in WT p53-expressing MCF7 cells (Figure 4A–C) and MEFs (Figure 4D,E). Together, these data suggest that, depending on the p53 status, NINJ2 may have opposing effects on cell growth and migration.

## 4. Discussion

In the current study, we discovered a novel feedback loop between NINJ2 and p53. We found that *NINJ2* is induced by wild-type p53 and can in turn regulate p53 mRNA translation (Figure 1 and Figure 2). Interestingly, the loss of NINJ2 inhibits cell growth and migration and decreases cellular senescence in WT p53-expressing cells (Figure 4), whereas the loss of NINJ2 promotes cell growth and migration in mutant p53-expressing cells (Figure 5). Together, these data indicate that the mutual regulation represents a negative feedback loop between NINJ2 and p53, and the loop may play a critical role in tumor suppression or tumor promotion depending on the genetic status of the p53 gene.

We showed that p53 mRNA translation is enhanced by the loss of NINJ2 (Figure 2E). However, the underlying mechanism remains to be elucidated. Indeed, several cell adhesion molecules have been found to regulate mRNA translation via different means. For example, α6β1-integrins were found to promote mRNA translation of the myelin basic protein by facilitating the release of mRNA from transport granules [41]. In addition, Neuroligin 3, a cell adhesion molecule, was found to mediate protein synthesis via activating the mammalian target of rapamycin (mTOR) signaling pathway [42]. Thus, it is possible that NINJ2 may associate with an RNA-binding protein and, subsequently, modulate p53 mRNA translation. It is also possible that NINJ2 participates in a signaling pathway, such as the mTOR pathway [43,44] known to modulate p53 translation. These possibilities will be examined in a future study to elucidate how NINJ2 mediates p53 mRNA translation. 

Our data indicate that targeting NINJ2 enhances wild-type p53 expression andsubsequently, cellular senescence and growth suppression (Figure 4). Thus, NINJ2 may be targeted for killing cancer cells with WT p53. Indeed, we showed previously that a peptide derived from the N-terminal extracellular domain of NINJ1 can enhance wild-type p53 expression [30]. Therefore, it will be interesting to determine whether a peptide derived from NINJ2 can elicit growth suppression via enhancing WT p53 expression. However, we would like to note that in our study, we also found that the knockout of NINJ2 leads to the enhanced expression of mutant p53 (Figure 5). While targeting NINJ2 can enhance wild-type p53 expression and elicit tumor suppression in WT p53-harboring cancer cells, targeting NINJ2 in mutant p53-harboring cancer cells may enhance tumor cell growth. Indeed, the TCGA database indicates that NINJ2 is over-expressed in ovarian serous cystadenocarcinoma and pancreatic adenocarcinoma, both of which are characterized by a high rate of p53 mutation [45,46,47]. As such, p53 status needs to be considered when developing strategies to target NINJ2 for cancer treatment. 

NINJ1 and NINJ2 belong to the Ninjurin family of cell adhesion molecules. However, compared with NINJ1, NINJ2 remains largely unexplored in terms of its biological function(s). Owing to their structural similarity, it is possible that NINJ2 may also participate in the signaling pathway, as does NINJ1. For example, NINJ1 was recently found to regulate plasma membrane rupture and promote pyroptosis [48,49], a programmed cell death that is originally found in macrophages. Notably, NINJ2 is highly expressed in hematopoietic cells including macrophages. Thus, it would be interesting to determine whether NINJ2 plays a role in pyroptosis. In addition, it would also be important to determine whether NINJ2 collaborates with or antagonizes NINJ1 in pyroptosis. Addressing these questions would not only help better understand the biological significance of NINJ2 but also lay a foundation for the development of targeting NINJ2 for disease management.

## Figures and Tables

**Figure 1 cancers-16-00229-f001:**
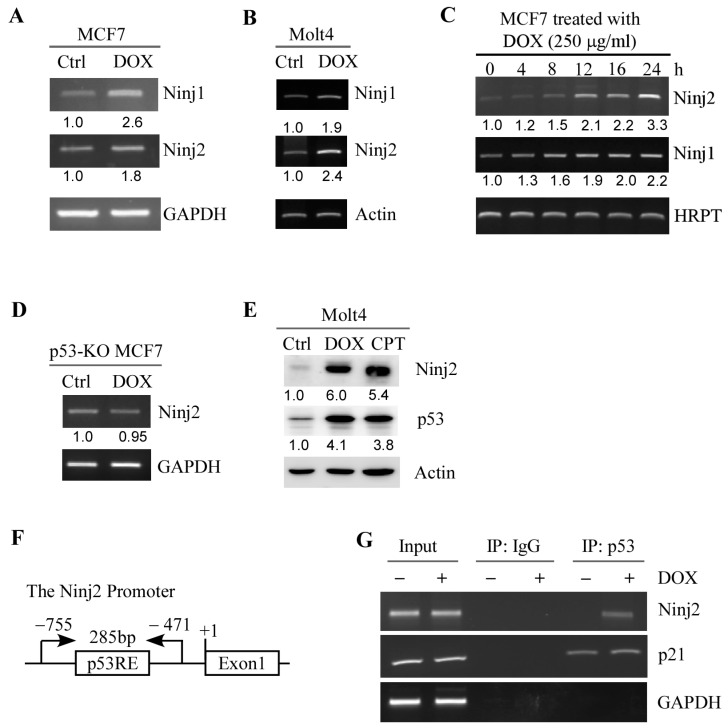
Ninj2 is induced by p53. (**A**,**B**) The levels of the *NINJ1*, *NINJ2* and *GAPDH* transcripts were measured by RT-PCR in MCF7 (**A**) and Molt4 (**B**) cells mock-treated or treated with doxorubicin (250 μg/mL) for 24 h. *GAPDH* was measured as an internal control. (**C**) The levels of *NINJ1, NINJ2* and *HRPT* were measured by RT-PCR in MCF7 cells treated with doxorubicin (250 μg/mL) for 0–24 h. *HRPT* was measured as an internal control. (**D**) The levels of *NINJ2* and *GAPDH* transcripts were measured in *p53-KO* MCF7 cells treated with or without doxorubicin. (**E**) The levels of NINJ2, p53, and Actin proteins were measured in Molt4 cells mock-treated or treated with doxorubicin (250 μg/mL) or camptothecin (250 μM) for 18 h. (**F**) Schematic representation of the *NINJ2* promoter and the locations of p53 response element (p53RE) and primers used for DNA-ChIP assay. (**G**) MCF7 cells were treated with or without doxorubicin, followed by ChIP assay to measure the binding of p53 to the NINJ2 and p21 promoter. The binding of p53 to the GAPDH promoter was used as a negative control.

**Figure 2 cancers-16-00229-f002:**
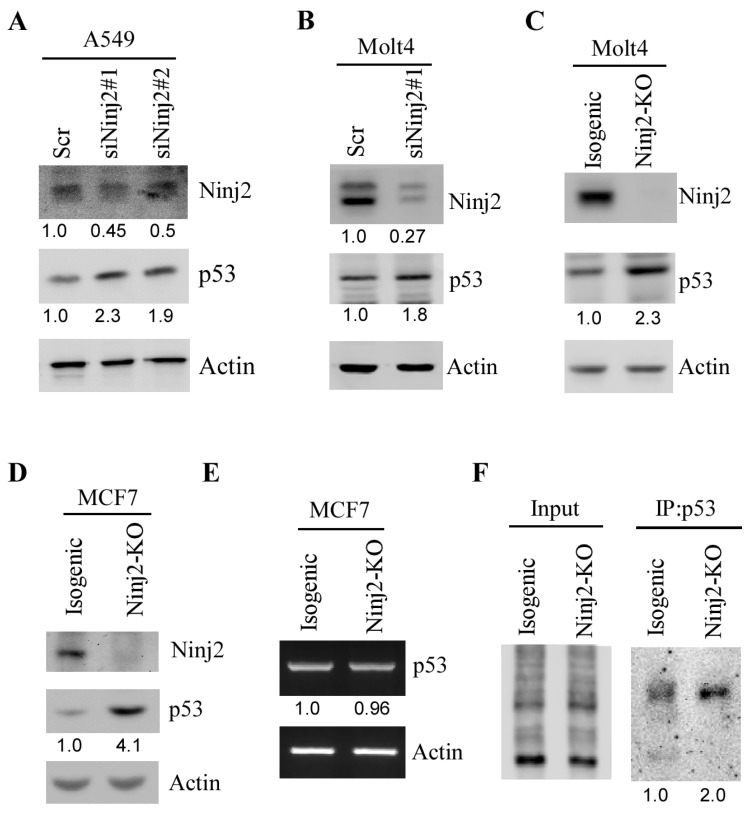
Ninj2 represses p53 expression via mRNA translation. (**A**,**B**) The levels of NINJ2, p53 and Actin proteins were measured by Western blot analysis in A549 (**A**) and Molt4 (**B**) cells transiently transfected with a scrambled siRNA or a siRNA against NINJ2 (siNINJ2#1 or #2) for 3 days. (**C**,**D**) The levels of NINJ2, p53 and actin proteins were measured in isogenic control and NINJ2-KO Molt4 (**C**) and MCF7 (**D**) cells. (**E**) The levels of p53 and actin transcripts were measured in isogenic control and NINJ2-KO MCF7 cells. (**F**) The level of newly synthesized p53 protein was measured in isogenic control and NINJ2-KO Molt4 cells by Click-iT metabolic assay according to the manufacturer’s manual.

**Figure 3 cancers-16-00229-f003:**
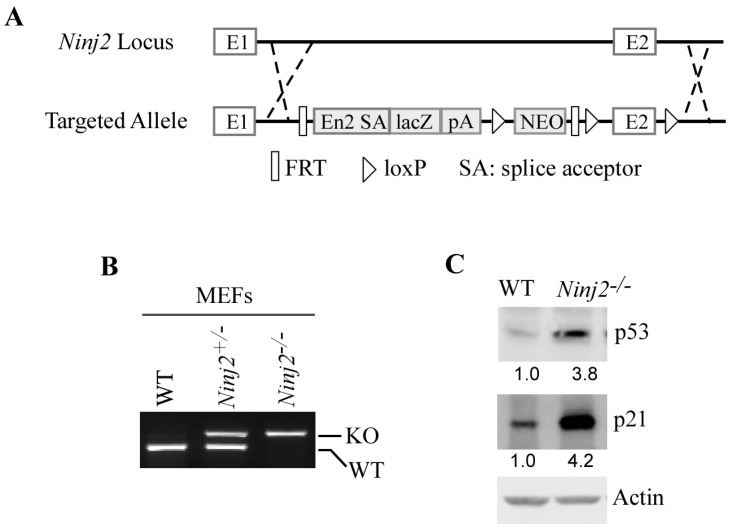
Loss of Ninj2 leads to increased expression of p53 accompanied by induction of p21 in MEFs. (**A**) The targeting strategy for generating *Ninj2*-KO mice. The Ninj2 transcript is trapped through the Engrailed-2 splice acceptor (En2 SA) element and truncated via the SV40 polyadenylation signal (pA). (**B**) Genotypes of WT, *Ninj2*^+/−^ and *Ninj2*^−/−^ MEFs were determined by PCR. (**C**) The levels of p53, p21 and actin proteins were measured in WT and *Ninj2*^−/−^ MEFs.

**Figure 4 cancers-16-00229-f004:**
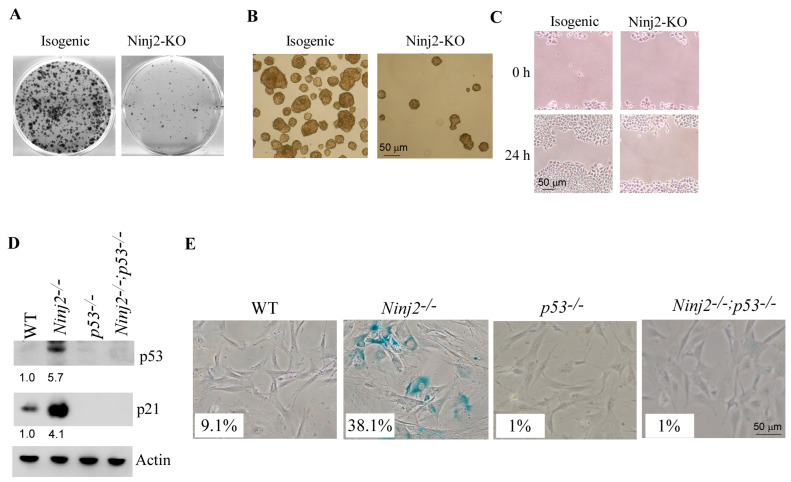
Loss of NINJ2 increases the ability of wild-type p53 to induce growth suppression, decrease cell migration and promote cellular senescence. (**A**–**C**) Isogenic control and *NINJ2*-KO MCF7 cells were used for colony formation assay (**A**), tumor sphere formation assay (**B**) and scratch wound healing assay (**C**). (**D**) The levels of p53, p21 and Actin proteins were measured in WT, *Ninj2*^−/−^, *p53*^−/−^ and *Ninj2*^−/−^; *p53*^−/−^ MEFs. (**E**) SA-β-Gal staining was carried out with WT, *Ninj2*^−/−^, *p53*^−/−^ and *Ninj2*^−/−^; *p53*^−/−^ MEFs.

**Figure 5 cancers-16-00229-f005:**
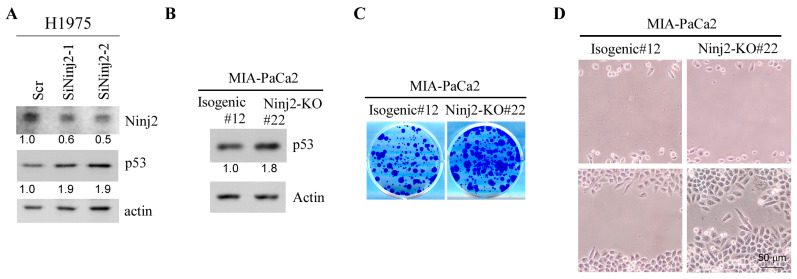
Loss of NINJ2 increases mutant p53 expression and promotes cell growth and migration. (**A**) The levels of NINJ2, mutant p53 and actin were measured in H1975 cells transiently transfected with a scrambled siRNA or an siRNA against Ninj2 for 3 days. (**B**) The levels of mutant p53-R248W and Actin were measured in isogenic control and *NINJ2-KO* MIA-PaCa2 cells. (**C**,**D**) Isogenic control and *NINJ2-KO* MIA-PaCa2 cells were used for colony formation assay (**B**) and scratch wound healing assay (**C**).

## Data Availability

Data were generated by the authors and included in the article.

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
