# Peer review of "Ninjurin 2, a Cell Adhesion Molecule and a Target of p53, Modulates Wild-Type p53 in Growth Suppression and Mutant p53 in Growth Promotion"

_cancers, 2024, doi:10.3390/cancers16010229_

Round 1

Reviewer 1 Report

Comments and Suggestions for Authors

The authors have identified nerve injury induced protein 2 (NINJ2) and p53 are mutually regulated at posttranslational level. The authors have used multiple assays cell lines and MEFs to show that NINJ2 is a p53-direct target and loss of p53 expression is stabilized via mRNA translation by NINJ2. More importantly, NINJ2 regulation of p53 expression leads to growth suppression and premature senescence in wild-type p53 cells, but NINJ2 induced mutant p53 expression promotes cell growth and migration in mutant p53-expressing cells. The authors have demonstrated that a mutual regulation mechanism between Ninj2 and p53 in cells. The work is sold and interesting. Only a few minor points need to address.

1.     In Figure 1 A,B,C,D, it will be better to have a bar graph and quantitation results from the PCR experiments.

2.     In Figure2A, two siRNA will be more convincing. Also in Figure 2B, 2C and 2D, the colonies of KO will be better.

3.     In Figure 3, are there any phenotypes in NINJ2 knockout mice related to higher p53 activity? For example, is spleen or thymus more sensitive to radiation in NINJ2 knockout mice?

4.     In Figure 5A, please quantitate the western blots to see the induction of p53 in knockout cells.

Author Response

Please see the attached file for reponses.

Reviewer 2 Report

Comments and Suggestions for Authors

This study try the clarify the  NINJ2 how to regulate the wild-type or mutated p53’s functions. The most of the experimental assays were with the reason meaning but there are several things should be verified or added as the following: 

1. Only one cell line with one hot spot mutation (MIA-PaCa2 cells with p53-R248W) to study the function of NINJ2 on p53 mutation is too preliminary. It should add another cell line with different hot spot mutation to study NINJ2 with the same response or not.

2. P21 can be regulated by non-p53 related pathway. But p21 showed almost no band in Figure 4D. So it should add the real time RT-PCR results with p21 specific primers to qualify the results again in.

3. All the microscopic figure should add the scale bar.

4. SA-β-Gal is almost all showed as SA- -Gal. It should correct all of them.

Round 2

Reviewer 2 Report

Comments and Suggestions for Authors

The revised version now can be accepted.